# Measurement of China’s Green Total Factor Productivity Introducing Human Capital Composition

**DOI:** 10.3390/ijerph192013563

**Published:** 2022-10-19

**Authors:** Can Cheng, Xiuwen Yu, Heng Hu, Zitian Su, Shangfeng Zhang

**Affiliations:** 1School of Statistics and Mathematics, Zhejiang Gongshang University, Hangzhou 310018, China; 2Modern Business Research Center, Zhejiang Gongshang University, Hangzhou 310018, China; 3Collaborative Innovation Center of Statistical Data Engineering, Technology and Application, Zhejiang Gongshang University, Hangzhou 310018, China

**Keywords:** green total factor productivity, human capital composition, SBM-GML index

## Abstract

In the face of resource and environmental problems caused by extensive economic development, China has put forward a green development strategy. Scientific measurement and analysis of green total factor productivity (GTFP) is of great significance for achieving high-quality economic development. By introducing the human capital composition, including education, health, scientific research, and training, this paper study adopts the Slack Based Measure-Global Malmquist-Luenberger (SBM-GML) index to re-measure the GTFP and its decomposition of 30 provinces, municipalities, and autonomous regions in China from 2000 to 2019. The results show that: (1) China’s GTFP has a fluctuating growth trend, with an average annual growth rate of 2.31%. (2) In terms of its decomposition, technical progress is the main force driving GTFP growth, with a contribution rate of 1.59%; the improvement of technical efficiency is a secondary driving force, with a contribution rate of 0.71%. (3) The measurement results of GTFP, considering the human capital composition, are generally higher than those without consideration, and the GTFP growth under the two modes shows a trend of “high in the east and low in the west”. The conclusions have enlightening significance for improving GTFP and the growth potential of the economy in China.

## 1. Introduction

Over the past 40 years of reform and opening-up, China’s economic development has made remarkable achievements, but this has been accompanied by the increasingly serious problems of resource consumption and environmental deterioration. A good ecological environment is the basic condition for regional sustainable development, and the problem of resources and environment will certainly restrict social-economic development. According to the World Bank, China’s economic losses due to pollution and environmental deterioration accounted for 10.51 percent of gross national income in 2008 [1]. In addition, the continuous deterioration of resource and environmental problems also cause great harm to the health of residents. The 2020 Report on Global Cancer Burden showed that China ranked first in cancer incidence and mortality in the world [2]. It can be seen that simply pursuing rapid economic growth is no longer suitable for China’s development conditions; thus, it is urgent to pursue the green transformation of production modes.

Against the background that resources and environment have increasingly become a hard constraint on economic growth, the Chinese government has set targets for energy conservation and emission reduction, and has taken corresponding measures. The Fifth Plenary Session of the 18th Communist Party of China (CPC) Central Committee took green development as one of the five development concepts, and put it into the “13th Five-Year Plan”. The report to the 19th National Congress of the CPC pointed out that high-quality economic development is to achieve the joined and coordinated development of economy, society, and ecological environment; pursing green development was placed into an extremely important position. The “14th Five-Year Plan” calls for accelerating green transformation of development modes. This leads to the question: How should the Chinese government scientifically evaluate the level of green development?

Economic history shows that total factor productivity (TFP) is a good proxy for the quality of economic growth among countries and regions, but the traditional TFP cannot reflect the true state of economic growth, because without considering the undesirable output of environment, it will give biased results to productivity measures [3,4]. GTFP has been used to measure high-quality economic development by comprehensively considering the degree of energy consumption and deterioration of the ecological environment [5]. Thus, how to measure GTFP has become an important basic research work. From the existing relevant research, there are mainly two research ideas: (1) introducing environmental pollution as an input factor into the production function [6,7,8]; (2) classifying environmental pollution as an undesired output [9,10,11]. However, most of these studies focus on the environmental factors themselves, and only consider labor quantity, ignoring the impact of human capital on GTFP. Human capital is an important source of technical progress and plays a key role in improving the efficiency and quality of economic growth. One essential positive externality of human capital development is that it promotes a greener future through energy conservation and improved energy efficiency [12,13]. Therefore, it is necessary to introduce human capital into the measurement of GTFP.

In recent years, some scholars have begun to study the relationship between human capital and GTFP in China, but there are still some gaps. First, most studies mainly use the level or years of education to measure human capital and ignore other factors that form human capital, such as health, training, and scientific research, which covers up the heterogeneity of human capital [14]. Second, in terms of measurement methods, existing research on human capital and GTFP has mostly adopted an SBM model and the Malmquist-Luenberger (ML) index based on undesired outputs. There are few studies on the comprehensive application of a SBM and the GML index, so the three key problems of slack variable, effective decision making unit (DMU) distinguishability, and inter-temporal comparability cannot be solved at the same time, which affects the robustness of measurement results [15,16]. Therefore, this study introduces human capital composition into the accounting framework of GTFP, adopts the SBM-GML index method to re-measure the GTFP, and 30 provincial samples of China from 2000 to 2019. The empirical results show that the estimated GTFP with the introduction of human capital composition is higher than that without the introduction.

The main contribution of this study is that the intrinsic attributes of human capital, including education, health, research, and training, are fully considered when measuring GTFP. It not only fills the gap in the research on the relationship between human capital and GTFP, but also provides a research reference for achieving the goals of green development and high-quality economic growth.

The remainder of this paper is organized as follows. Section 2 presents a literature review. Section 3 articulates the research methodology, including the research model, variable measurements, and data description. The empirical results and discussion are provided in Section 4. Finally, Section 5 summarizes the study.

## 2. Literature Review

Based on the research objective, this study mainly sorts the relevant literature into three features: the measurement of human capital, the measurement of GTFP, and the relationship between human capital and GTFP.

### 2.1. Measurement of Human Capital

The concept of human capital can be traced back to the late 1950s. Human capital is the talent, knowledge, technology, and health that condense on workers through education, training, medical treatment, migration, etc., and are then manifested by specific labor [17]. Thus, high-quality human capital can achieve higher labor productivity. As human capital is a kind of intangible asset, there is no unified measurement method in academia. At present, there are three commonly used methods to measure human capital: namely, cost method, education index method, and income method. Based on the investment cost, the cost method determines the current value level of human capital according to the accumulated input cost in the process of human capital accumulation [18]. Later, many experts extensively studied and expanded the cost method, which further developed the idea of cost method and gradually formed the perpetual inventory method, which was widely applied [19,20,21,22]. The education index method believes that the formation of human capital is centered on the accumulation of education, and the index of “years of education” is mostly used to estimate human capital [23,24], and some scholars choose other indicators, such as literacy and the number of graduates [25,26]. The income method determines the current value level of human capital based on the total present value of income returns that human capital can obtain during the whole service period [27,28].

### 2.2. Measurement of GTFP

There have long been studies that incorporate resources and environment into the production function to measure the level of sustainable development [29]. The ML index based on directional distance function (DDF) had been used to measure the TFP, including pollution emissions as an undesired output, and reasonably fitted the effect of pollution emissions on economic growth for the first time from the method, which obtained the true GTFP [9]. Since then, many scholars have adopted the ML index to measure GTFP [30,31]. However, the traditional ML index has the problematic potential to provide biased results when existing non-zero slacks. In order to solve this problem, some scholars proposed a method to measure the non-radial and non-oriented slacks, and developed a non-radial and non-oriented DEA approach (SBM model) based on this [32,33]. Since the SBM model has advantages on efficiency analysis from multiple inputs and multiple outputs, many studies have applied this model and the ML index to measure GTFP [34,35,36]. In addition, the ML index does not consider inter-temporal DEA, which will cause problems such as infeasible solutions and non-transitiveness of measurement results. In order to solve the above problems, the GML index had been used to measure spatial convergence of GTFP in China’s primary provinces along its Belt and Road Initiative, differences of GTFP in 163 countries (or regions) around the world, and convergence of GTFP in China’s service industry [5,37,38,39].

### 2.3. Relationship between Human Capital and GTFP

At the level of theoretical research, the routes of human capital’s impact on GTFP fall into four groups: (1) spillover on technical progress; (2) spillover on knowledge; (3) matching the upgrading of industrial structure; (4) enhancing environmental awareness [40,41,42,43,44,45,46,47]. At the level of empirical research, with increasing attention paid to GTFP, the relationship between human capital and GTFP has aroused the research interest of scholars. Tan et al. (2016) considered the influence of human capital when verifying the spatial learning effect of provincial GTFP, and the empirical results showed that the input of human capital was conducive to the improvement of the change in green technology efficiency [48]. Zhu and Wang (2019) studied the influencing factors of provincial GTFP by constructing a systematic Generalized Method of Moments (GMM) model, and the regression results showed that human capital contributed to the double effects of green technology progress and green technology efficiency regression [49]. Yin and Li (2019), based on the inter-provincial panel data from 2008 to 2017 and using Data Envelopment Analysis-Explore Spatial Data Analysis (DEA-ESDA) method, concluded that human capital level significantly restricted the growth of GTFP, and human capital level had a significant negative impact on the improvement of technical efficiency [50]. Zhang and Hu (2021) considered the Yangtze River Delta region as the research object to explore the spatial effect of innovative human capital on GTFP. The empirical results showed that the increase in innovative human capital investment will hinder the improvement of GTFP level, and there may be an “island effect” of innovative human capital [51]. Su and Zhou (2021) found that compared with scientific and technological innovation, human capital has a stronger driving force for the improvement of GTFP [52]. Based on literature research results, human capital may have a positive or negative relationship with GTFP or green development efficiency.

Based on relevant domestic and foreign literature, it can be found that there are two problems in the research on the impact of human capital on GTFP: (1) Education indicator method is most favored by scholars because of the availability of data. Most studies use years of education to estimate human capital stock, ignoring other factors that form human capital, such as health, training, scientific research, etc., which covers up the heterogeneity of human capital. (2) Although existing literatures have begun to study the relationship between human capital and GTFP, there are few studies on the comprehensive application of SBM and the GML index, which makes the estimation results weak. Based on this, this study will focus on the impact of human capital composition on regional GTFP by adopting the SBM-GML index method.

## 3. Model and Data Description

### 3.1. SBM-GML Model

At the level of theoretical research, there are two main methods to measure GTFP: parametric method and non-parametric method. The former needs to set a specific form of production function, which means that a series of assumptions must be satisfied; otherwise, it will easily lead to the deviation of estimation results. The latter, which is represented by the Data Envelopment Analysis (DEA) and the Stochastic Frontier Analysis (SFA), does not need to set a specific functional form, avoiding strong theoretical constraints, so it is more suitable for measuring GTFP [34]. Compared with the SFA, the DEA considers the linear programming idea of multi-input and multi-output, which makes it easier to compare the distance between the DMU and the technical progress frontier, so it is favored by many researchers. To avoid the defects that the ML index method cannot reflect the long-term growth trend of productivity, and may have no feasible solution for linear programming, this study introduces the SBM-GML model based on DEA method.

#### 3.1.1. Global Production Possibility Set

Consider a province as a DMU, assuming that each DMU uses *M* inputs, denotes as $x=\left(x_{1},x_{2},\cdots x_{m}\right)\in R_{+}^{M}$, generates N kinds of desired outputs $y=\left(y_{1},y_{2},\cdots y_{n}\right)\cdots R_{+}^{N}$, and *J* kinds of undesired outputs $b=\left(b_{1},b_{2},\cdots ,b_{m}\right)\in R_{+}^{J}$. So, the input-output value of *k* province in period *t* can be expressed as $\left(x^{k,t},y^{k,t},b^{k,t}\right)$. Closed set and bounded set, free disposability of inputs and desired outputs, joint weak disposability of undesired outputs, axiom of zero combination of outputs are the assumptions that need to be satisfied in the production possibility set, from which the current production feasibility set can be constructed: $P^{t}\left(x^{t}\right)=\left\{\left(y^{t},\;b^{t}\right):x^{t}\right\}$.

However, $P^{t}\left(x^{t}\right)$ is a set of production technology constructed by the current production technology in *t* period, which may lead to the conclusion of “regression of technology”, and the result of efficiency measurement may be biased. Oh (2010) improved the production technology set of the current period, and set the global production technology set by using the observed data throughout the whole period of the production set [37].
(1)$$P^{G}(x)=\{\begin{array}{c} (y^{t},b^{t})|\sum_{t=1}^{T}\sum_{k=1}^{K}\lambda_{k}^{t}y_{kn}^{t}\geq y_{n}^{t},n=1,2,\cdots ,N\\ \sum_{t=1}^{T}\sum_{k=1}^{K}\lambda_{k}^{t}x_{km}^{t}\geq x_{m}^{t},m=1,2,\cdots ,M\\ \sum_{t=1}^{T}\sum_{k=1}^{K}\lambda_{k}^{t}b_{kj}^{t}\geq b_{j}^{t},j=1,2,\cdots ,J\\ \sum_{k=1}^{K}\lambda_{k}^{t}=1,\lambda_{k}^{t}\geq 0,k=1,2,\cdots ,K \end{array}$$
where $\lambda_{k}^{t}$ denotes the weight of input-output value of *k* province in *t* period. The global production possibility set considers the production technology level of all periods; namely, $P^{G}\left(x\right)=P^{1}\left(x^{1}\right)\cup^{\;}P^{2}\left(x^{2}\right)\cup^{\;}\cdots \cup^{\;}P^{T}\left(x^{T}\right)$, defined as the union of all production technology sets in the current period, which enhances the comparability of efficiency levels among decision-making units in different periods.

#### 3.1.2. SBM Model

Since the DDF can treat the desired output and the undesired output differently and obtain the optimal solution of the production possibility set, it is widely used in efficiency evaluation problems involving “bad” outputs. The function of directional distance is expressed as D(x,y,b;g)=max{β:(y,b)+βg∈P(x)}, where, directional vector is g=(y,b), β is a DDF that seeks to maximize “good” outputs and minimize “bad” outputs. The measurement of DDF for ineffective DMUs only includes the proportional change of input and output variables, but does not consider the improvement of the non-zero slack term.

In order to overcome the limitations of DDF, this study adopts the SBM model.
(2)$$\begin{array}{l} S_{V}^{G}(x_{k}^{t},y_{k}^{t},b_{k}^{t};g^{x},g^{y},g^{b})=\underset{s^{x},s^{y},s^{b}}{\max}\frac{\frac{1}{M}\sum_{m=1}^{M}\frac{S_{m}^{x}}{g_{m}^{x}}+\frac{1}{N+J}(\sum_{n=1}^{N}\frac{S_{n}^{y}}{g_{n}^{y}}+\sum_{j=1}^{J}\frac{S_{j}^{b}}{g_{j}^{b}})}{2}\\ s.t.\;\sum_{t=1}^{T}\sum_{k=1}^{K}\lambda_{k}^{t}x_{km}^{t}+S_{m}^{x}=x_{mK}^{t},m=1,2,\cdots ,M\\ \sum_{t=1}^{T}\sum_{k=1}^{K}\lambda_{k}^{t}y_{kn}^{t}-S_{n}^{y}=y_{nK}^{t},n=1,2,\cdots ,N\\ \sum_{t=1}^{T}\sum_{k=1}^{K}\lambda_{k}^{t}b_{kj}^{t}+S_{j}^{b}=b_{jK}^{t},j=1,2,\cdots ,J\\ \sum_{k=1}^{K}\lambda_{k}^{t}=1.\lambda_{k}^{t}\geq 0,k=1,2,\cdots ,K\\ S_{m}^{x}\geq 0,S_{n}^{y}\geq 0,S_{j}^{b}\geq 0 \end{array}$$
where the vector $\left(x_{k}^{t},y_{k}^{t},b_{k}^{t}\right)$ indicates DMU*_k_*’s input, desired output, and undesired output vector in *t* period. $\left(g^{x},g^{y},g^{b}\right)$ are positive directional vectors that contract inputs, undesired outputs, and expand desired outputs. $\left(S^{x},S^{y},S^{b}\right)$ denotes the vectors of input, desired output, and undesired output slack, which have the same units of measurement as $\left(g^{x},g^{y},g^{b}\right)$. The objective of the first equation in (2) is to maximize the sum of average input inefficiency and average output inefficiency.

#### 3.1.3. GML Index

The GML index takes the sum of all periods as reference set, which has advantages on inter-temporal comparability. According to Oh (2010), it can be expressed as:(3)$$GML_{t}^{t+1}=\frac{1+S_{V}^{G}(x^{t},y^{t},b^{t};g)}{1+S_{V}^{G}(x^{t+1},y^{t+1},b^{t+1};g)}$$
where $GML_{t}^{t+1}$ denotes the GTFP in *t* + 1 period. If $GML_{t}^{t+1}>1$, indicating that the GTFP increases from *t* to *t* + 1; If $GML_{t}^{t+1}=1$, indicating that the GTFP is constant from *t* to *t* + 1; If $GML_{t}^{t+1}<1$, indicating that GTFP is falling from *t* to *t* + 1.

Since GML index takes the sum of each period as the reference set, and the two adjacent periods refer to the same global production frontier without crossover of production frontier. Therefore, the GML index can only be decomposed into the global efficiency change index (GEFFCH) and the global technology progress change index (GTECH), which cannot be further subdivided. This paper will explore the sources of changes in China’s GTFP through decomposition of the GML index, and the specific decomposition is as follows:(4)$$\begin{array}{ll} GML_{t}^{t+1} & =\frac{1+S_{V}^{t}(x^{t},y^{t},b^{t};g)}{1+S_{V}^{t+1}(x^{t+1},y^{t+1},b^{t+1};g)}\times \frac{\left[1+S_{V}^{G}(x^{t},y^{t},b^{t};g)]/[1+S_{V}^{t}(x^{t},y^{t},b^{t};g)\right]}{\left[1+S_{V}^{G}(x^{t+1},y^{t+1},b^{t+1};g)]/[1+S_{V}^{t+1}(x^{t+1},y^{t+1},b^{t+1};g)\right]}\\ & =GEFFCH_{t}^{t+1}\times GTECH_{t}^{t+1} \end{array}$$
where $GEFFCH_{t}^{t+1}$, $GTECH_{t}^{t+1}$ respectively denotes changes in efficiency and technological progress from *t* to *t* + 1. If $GEFFCH_{t}^{t+1}>1$, meaning production efficiency grows from *t* to *t* + 1; If $GEFFCH_{t}^{t+1}=1$, meaning production efficiency is constant from *t* to *t* + 1; If $GEFFCH_{t}^{t+1}<1$, meaning production efficiency is falling from *t* to *t* + 1. If $GTECH_{t}^{t+1}>1$, meaning progress in production technology from *t* to *t* + 1; If $GTECH_{t}^{t+1}=1$, meaning no changes in production technology from *t* to *t* + 1; If $GTECH_{t}^{t+1}<1$, meaning regression in production technology from *t* to *t* + 1.

### 3.2. Selection of Variables 

In this paper, the composition of GTFP indicators is divided into input variables and output variables. Factors of production are generally divided into labor, capital, natural resources, and entrepreneurial talent. Entrepreneurial talent is often used to examine high-quality economic development from the micro level [53,54]; this paper measures China’s GTFP from the macro level. Therefore, we take labor, capital, and energy as input variables [55,56,57]. Since the measurement of GTFP considers both increases in desired output and decreases in undesired output, output variables should be divided into two aspects: desired output and undesired output.

#### 3.2.1. Input Variables 


Labor. Since this paper considers human capital composition, it is necessary to distinguish between labor quantity and quality, and the quality includes education, health, scientific research, and training. (1) Labor quantity. Similar to most studies, this paper uses the total number of employees engaged in social labor and paid at year-end to represent labor quantity. (2) Education. This paper uses average years of education as a proxy for education human capital, which is generally expressed by the product of the proportion of educated labor at all levels and education years. According to Thomas et al. (2001), we divide the education level into seven grades: illiterate, primary school, junior high school, high school and secondary vocational school, college, undergraduate, postgraduate, and above [58]. In this way, the calculation formula of average years of education can be expressed as $H_{edu}=\overset{7}{\underset{i}{\sum }}l_{i}h_{i}$, where *i* is the education level, $l_{i}$ denotes proportion of educated labor at all levels, $h_{i}$ denotes cumulative years of education corresponding to each education level, which is respectively set as 0, 6, 9, 12, 15, 16, and 20 (Table 1) [59,60]. (3) Health. A higher proportion of government health expenditure can cultivate more healthy human capital, so we use the proportion of total health expenditure in gross domestic product (GDP) as a proxy for health. (4) Scientific research. The research and development (R&D) personnel full-time equivalent is an internationally accepted indicator for comparing scientific and technical manpower inputs, so it is taken as the proxy of scientific research human capital in this paper. (5) Training. Considering the availability of data, based on the “Decision” promulgated in 2002, this paper adopts 1.5% of the employee’s gross salary to represent the human capital used for training. Capital. Capital input is represented by fixed capital stock, which is calculated by the perpetual inventory method: $K_{i,t}=\left(1-\delta \right)K_{i,t-1}+I_{i,t}/d_{i,t}$. $K_{i,t}$ denotes the physical capital stock of region *i* in period *t*, $I_{i,t}$ denotes the total investment in fixed assets at the current price of region *i* in period *t*, $d_{i,t}$ denotes the price index of the total investment in fixed assets. δ is the depreciation rate, which is set at 9.6% [61].Energy. Many scholars have taken energy consumption as a proxy for energy input, consistent with most literature, this paper adopted “total energy consumption” to represent energy input [11,62].


#### 3.2.2. Output Variables

Desired output. This paper chooses regional GDP to represent the desired output. The annual regional GDP recorded in the Statistical Yearbook is estimated at current prices, but inflation will cause price changes, so it is necessary to eliminate the price factor. This paper takes 1978 as the base period and uses the GDP deflator to subtract the original data for obtaining the GDP per year, expressed at constant prices in 1978.Undesired output. Considering the current industrial three wastes (exhaust gas, waste water, and solid waste) in the process of Chinese industrialization, this paper adopts three types of domestic and industrial pollutant emissions as the proxy variable of undesired output: (1) Chemical oxygen demand (COD) emissions; (2) Sulfur dioxide (SO_2_) emissions; (3) Solid waste emissions. Due to the lack of emission data for solid waste, we take the production of solid waste as a substitute to obtain effective and robust econometric regression results.

The specific setting of indicators selected in this paper are shown in Table 2.

### 3.3. Data Source and Description

This paper takes 30 Chinese provinces (municipalities and autonomous regions) from 2000 to 2019 as the research sample (due to lack of data, Tibet, Hong Kong, Macao, and Taiwan are excluded). The full text data comes from the China Statistics Yearbook, China Health Statistics Yearbook, China Labor Statistics Yearbook, China Energy Statistics Yearbook, China Education Statistics Yearbook, Provincial Statistical Yearbook, and Easy Professional Superior (EPS) data platform. The exchange rate is derived from the annual average exchange rate of RMB against the US dollar by the National Bureau of Statistics. For missing data in some years, we use exponential smoothing to fill in.

Table 3 reports the descriptive statistics of the variables. It can be seen that there is a large gap in R&D personnel full-time equivalent between provinces, with the standard deviation as high as 109,321.5902, indicating that the scale of the scientific and technological personnel team is expanding, and the distribution of scientific and technological talents is seriously skewed. The ratio of maximum to minimum of training human capital is 21.444, and the standard deviation is close to the median. The absolute difference between average years of education and the proportion of total health expenditure in GDP is not large, but the relative difference is slightly larger. There is a large difference in capital input at the provincial level, and the provinces with a high fixed capital stock are more likely to absorb and accumulate capital, which leads to factor agglomeration. With the increasingly convenient transportation, labor transfer has been promoted, and labor mobility between provinces has been gradually enhanced. Economic development cannot be separated from energy input, so its distribution is relatively concentrated. At the same time, the standard deviation of desired output is greater than the mean value, while the maximum value of undesired output is more than 100 times of minimum value, indicating that the development mode of economic growth in most provinces is “extensive” at the cost of sacrificing environment. Therefore, there are large differences in the economic development level of China’s 30 provinces from 2000 to 2019, mainly reflected in two aspects: the economic scale and growth rate of each province, and the impact of economic development on the environment.

## 4. Empirical Results and Discussion

### 4.1. Empirical Results

With the introduction of human capital composition, this paper uses the SBM-GML index model and MaxDEA Pro6 software to measure the GTFP and its decomposition of 30 Chinese provinces and cities during 2000–2019, and analyzes its evolution characteristics from the time dimension and space dimension, respectively. Further, this paper makes a comparative analysis of the GTFP with the introduction of human capital and the traditional GTFP.

#### 4.1.1. Temporal Variation Characteristics

The numerical results of the temporal variation of GTFP and its decomposition show that the GTFP shows a fluctuating growth trend, with an average annual growth rate of 2.31%, and technological progress (1.59%) is the main driving force, while technical efficiency improvement (0.71%) is the secondary driving force. This is different from the research conclusion obtained by Zheng and Hu (2006), based on 30 provinces and regions in the 1990s; the possible reasons for this difference include time series differences, the selection of input-output factors, and calculation methods [63]. From the perspective of index decomposition: first, GEFFCH decreases in a fluctuating manner; second, GTECH fluctuates slightly from 2000 to 2015, and the fluctuation amplitude became larger after 2015; third, the change trend of GTFP and GTECH is basically consistent, indicating that the growth of GTFP and technological progress are inseparable in the context of health, education, scientific research, and training human capital factors.

Figure 1 shows the changing trend of the temporal variation of GTFP and its decomposition. It can be seen that there are three periods in which GTFP is in the growth stage, namely 2000–2007, 2009–2011, and 2013–2019, but the sources of GTFP growth in each period are different. Among them, the improvement of technical efficiency promoted the growth of GTFP in 2000–2001, 2003–2004, 2010–2012, and 2016–2017, while the years of 2001–2003, 2013–2016, and 2017–2019 are the result of technological progress. In addition, it is the joint effect of technical efficiency, improvement, and technological progress. From 2000 to 2007, China joined the World Trade Organization (WTO) and gradually improved the level of opening up. With the rapid development of international trade and foreign direct investment, China has played an important role in introducing and absorbing advanced technology from abroad. However, during 2000–2001 and 2003–2004, the level of technological progress declined, which may be because the relative price of input factors changed greatly as economic reform entered the critical stage, which adversely affected the choice of input factors. In addition, the strategy of “revitalizing the old industrial base in northeast China” in 2003 also caused a certain negative impact on technological progress. In 2008, the global financial crisis broke out, which severely affected import, export, and foreign investment. Private enterprises were also hit by financing difficulties, leading to technological level regression. GTFP decreased from 2011 to 2013, which may be caused by the fact that some provinces and cities in China have not yet got rid of the development model of high input, low efficiency, and high pollution.

#### 4.1.2. Regional Distribution Characteristics

In order to better observe the spatial distribution characteristics of GTFP among 30 Chinese provinces and cities, this paper further divides them into three major regions for measurement: eastern, central, and western. The specific results are shown in Table 4.

From the geometric mean of GTFP, the eastern, central, and western regions have achieved GTFP growth of 1.0263, 1.0229 and 1.0189, respectively. From the perspective of efficiency variation factors, the central and western region are generally higher than the eastern region, and the regions with efficiency values less than or equal to 1 are mainly distributed in the eastern regions (Beijing, Shanghai, Jiangsu, Zhejiang, Fujian, Shandong, Guangdong, and Hainan, all equal to 1). There are 1 or 2 provinces in central and western regions whose efficiency value is less than or equal to 1, respectively (Jiangxi in central China, Gansu and Xinjiang in western region). Among them, nine provinces, municipalities, and autonomous regions, including Beijing, Shanghai, Jiangsu, Zhejiang, Fujian, Shandong, Guangdong, Hainan, and Xinjiang, have a green technology efficiency value of 1, indicating that it has a neutral effect on efficiency improvement. The value of green technology efficiency in Jiangxi was 0.9973, and that in Gansu was 0.9964, showing an inhibition effect of nearly 0.4%. In terms of technological change factors, the eastern region is higher than the central region and the western region. The geometric mean of the technology progress index in the eastern region is 1.0221, followed by the central region (1.0124) and the lowest in the western region (1.0113).

#### 4.1.3. Compared with the Measurement Results without Human Capital

Comparison of GTFP in Two Modes.

At the national level, when human capital was included, the national average GTFP increased by 2.308% from 2000 to 2019. While, when human capital was not included, the national average GTFP increased by 0.87%. Therefore, the absence of human capital underestimates GTFP.

Figure 2 shows the measurement results of China’s GTFP under the two modes. It can be seen that the calculation results of GTFP are different, but the change trend of the two is basically the same. Taking 2009 as the segmentation point, from 2000 to 2009, the calculation results of the two are similar with slight differences. From 2009 to 2019, there was a significant difference between the two. The difference in GTFP reached 6.5% in 2015–2016 and 0.14% in 2012–2013. This shows that since 2009, China has paid more attention to human capital, attached greater importance to education and health, and led the transformation of labor and capital-intensive industries to knowledge- and technology-intensive industries.

From the regional level (Table 5), regardless of whether human capital factor input is considered, the growth of GTFP is the highest in the eastern region, followed by the central region, and the lowest in the western region. The central region has a good industrial base and rich human resources, but it is not dominant in capital, technology, and management. There are few emerging industries introduced by industrial transfer from the eastern region, but they bring more industrial pollution. The economic foundation of the western region is relatively weak. The implementation of the western development strategy has promoted the rapid economic development of the western region. The talent development has made the rationalization of the distribution of human resources and the foundation of talent team construction continuously strengthened, and gradually narrowed the gap with the eastern region.

At the level of provinces, if human capital is not included, GTFP of four provinces is less than 1 (Guangxi in the east, Heilongjiang in the central, Qinghai and Xinjiang in the west). When human capital is included, the GTFP of all provinces is greater than 1. As can be seen from Figure 3, the growth of GTFP based on human capital composition in the three major regions is greater than that without human capital composition, and they all show a situation of “high in the east and low in the west”. The main reason is that the economically active eastern region relies on its advantages and has produced a siphon effect on the central and western regions, which also reflects the fact that China’s unbalanced regional distribution of human capital. 

2.Comparison of the Decomposition of GTFP in Two Modes.

Figure 4 and Figure 5 show the time series comparison results of technical efficiency changes and technological progress decomposition terms of GTFP under different modes. It can be seen that the measurement results of the technical efficiency index incorporating human capital are lower than the traditional model in some periods: 2001–2002, 2015–2016, and 2017–2019. The measured results of the technology progress index incorporating human capital from 2000 to 2013 were lower than those of the traditional model, while the measured results of the technology progress index incorporating human capital from 2013 to 2019 were higher than those of the traditional model, and the gap showed a widening trend compared with the previous period.

According to the numerical results of the decomposed items (Table 6), when human capital is considered, the growth of the national average GTFP during 2000–2019 presents a “dual drive” growth mode of technical efficiency and technological progress. Among them, technological progress plays a prominent role in promoting the growth of GTFP (1.59%), which is the main driving force for the growth of GTFP. Efficiency improvement is also conducive to the growth of GTFP. The growth of GTFP in the eastern and western regions mainly depends on technological progress, and the contribution of efficiency improvement is relatively small. The contribution of technical efficiency improvement and technological progress to green all-purpose productivity growth in central regions is similar. Under the traditional mode, the national average GTFP growth from 2000 to 2019 mainly depends on technological progress, and technical efficiency does not play a role, which is also the case in the eastern, central, and western regions. The traditional technology progress index is smaller than the technology progress index considering human capital factors at the national level, eastern, central, and western level, indicating that China’s human capital development is effective. There is a commonality in the two modes; that is, technological progress is the main factor promoting the growth of GTFP.

Figure 6 shows the technical efficiency measurement results of each province (city, autonomous region) in China from 2000 to 2019. It can be seen that the technical efficiency index obtained by including human capital composition is equal to that obtained by traditional mode in the six places of Beijing, Shanghai, Fujian, Guangdong, Hunan, and Chongqing, while the technical efficiency index calculated by including human capital composition is larger than that calculated by traditional mode in other places. Among them, the technical efficiency index of Hebei, Guangxi, Inner Mongolia, Jilin, Heilongjiang, Henan, and Qinghai has a large gap between the two modes, which means that the technical efficiency can be greatly improved by raising the level of human capital in these provinces and cities.

Figure 7 shows the measurement results of technology progress in China at the province (city and autonomous region) level from 2000 to 2019. It can be seen that most provinces (cities and autonomous regions) have similar technological progress decomposition of GTFP. There is a large difference between Shandong and Xinjiang, indicating that human capital composition has a higher improvement effect on technical efficiency than technological progress. 

### 4.2. Discussion

It is natural for China, a developing country, to realize its “growth miracle” while bringing about negative problems, such as environmental pollution and resource waste. The problem of resources and environment cannot be ignored. Therefore, the “13th Five-Year Plan”, for the first time, incorporates the distinctive green development concept into the national Five-Year plan, and green development is a new driving force for economic growth. Human capital can promote productivity growth by absorbing and applying existing technologies or innovative technologies. It not only reflects the quality of the labor force, but is also an important component of technology absorption and innovation capacity. The relationship between human capital and economic development is complementary to each other and promotes each other. With the continuous improvement of labor quality, human capital also gradually develops into a new advantage of our economic growth. Based on the above, this paper used the SBM-GML index to measure China’s GTFP from 2000 to 2019, with the introduction of human capital composition. The results show that whether human capital is considered or not, China’s GTFP is increasing. The GTFP measured without considering human capital composition was lower. 

Compared with the existing literature, the second conclusion is the opposite [64]. There are two possible reasons for the contradiction. First, the time span of the sample is different. They selected data from 1990 to 2004, while the time span of this paper is from 2000 to 2019. The development of human capital has its own endogenous dynamic development and accumulation process, so the long-term accumulation of human capital is an important driving force for sustained economic growth. Second, the factor indicators considered are different. They characterized human capital by the average years of education, without taking undesired output into account. This study considered not only education, but also health, scientific research and training, and environmental pollution.

This study re-measured China’s GTFP from a new perspective of human capital composition, which can not only enrich the research content of GTFP, but also focus the attention to human capital, an important immaterial factor. However, there are still some shortcomings in this study, which can be further studied from the following approaches in the future: (1) Considering the data availability, this study only used COD emissions, SO_2_ emissions, and solid waste production to measure the undesired output, while the main pollutants stipulated in the “14th Five-Year Plan” for energy conservation and emission reduction include COD, nitrogen oxide, ammonia nitrogen, and volatile organic compounds. Therefore, it is necessary to expand the use of pollutant indicators. (2) This study measured human capital from the perspectives of education, scientific research, health, and training, but it is still a static study, and the dynamic migration of human capital could be further introduced in the future. (3) This paper measured China’s GTFP including human capital composition at the provincial spatial scale, which can be extended to the city level and enterprise level in the future.

## 5. Conclusions

The research significance of this paper lies in using the SBM-GML index to re-measure China’s GTFP at the province (city, autonomous region) level by introducing human capital composition. Our findings have valuable implications for promoting the high-quality development of China’s economy.
The GTFP measured based on human capital composition shows a fluctuating growth trend, with an average annual growth rate of 2.31%, which indicates that the efficiency of green growth has been improved across the country, the growth momentum is sufficient, and the economic growth mode is sustainable. Therefore, the improvement of human capital is an important means by which to achieve green economic development, and it is necessary to further increase the investment in human capital. The government should continue to strengthen the investment in education and the construction of medical and health security to ensure the necessary foundation for the improvement of human capital.From the decomposition of GTFP, technological progress (1.59%) is the main driving force, and technical efficiency improvement (0.71%) is the secondary driving force; that is, the national average GTFP growth from 2000 to 2019 showed a joint promotion of “dual-drive” growth model. This means that, with the improvement of human capital in education, health, scientific research, and training, the ability for scientific and technological innovation is enhanced, and technological progress is rapidly developing.The measurement results of China’s GTFP based on human capital composition are generally higher than that without human capital composition, and the growth of GTFP in both models presents a trend of “high in the east and low in the west”. Therefore, regions with high level of GTFP should play a leading role in promoting regional collaborative development by weakening the “siphon effect” between regions, emphasizing the radiation of human capital from “center” to “edge”, and strengthening green technology exchange and cooperation. In terms of decomposition, technological progress is the main factor promoting the growth of GTFP, while efficiency improvement has no positive effect on the GTFP without human capital.


## Figures and Tables

**Figure 1 ijerph-19-13563-f001:**
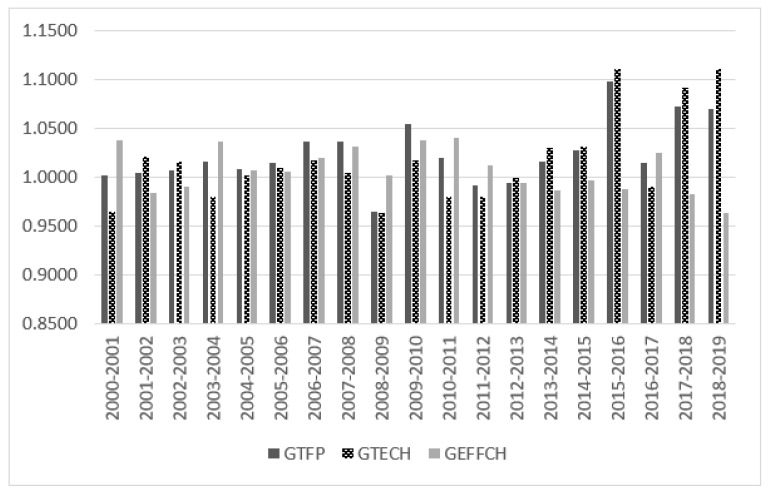
GTFP and its decomposition based on human capital composition.

**Figure 2 ijerph-19-13563-f002:**
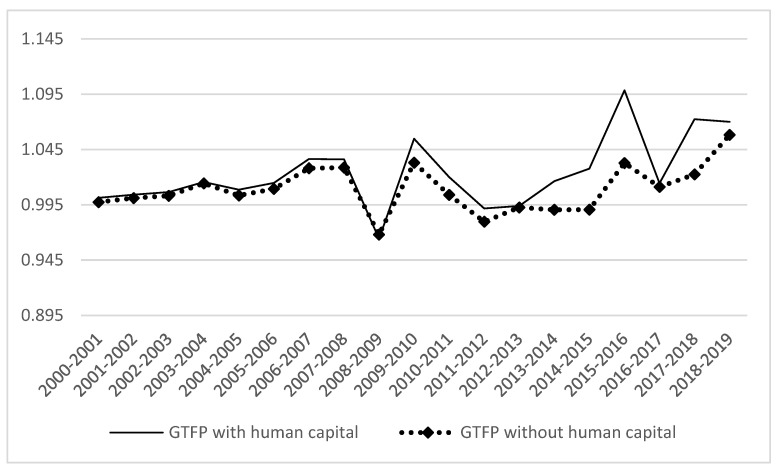
Comparison of GTFP in two modes.

**Figure 3 ijerph-19-13563-f003:**
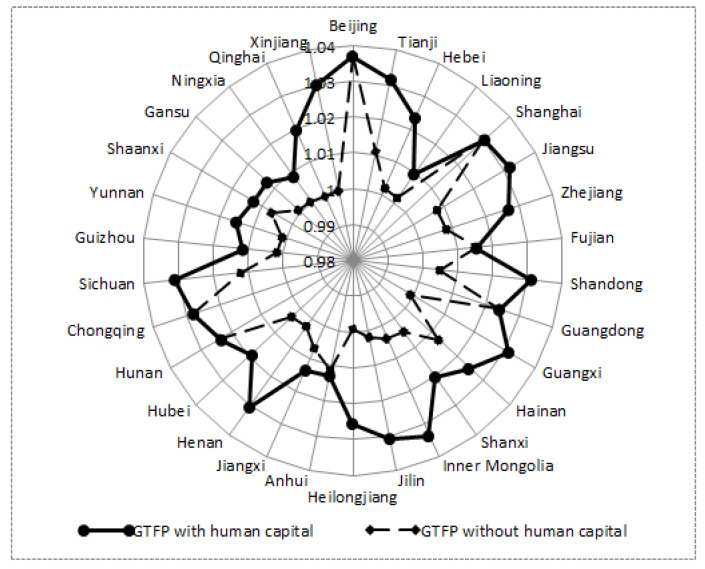
Comparison of GTFP among provinces and cities.

**Figure 4 ijerph-19-13563-f004:**
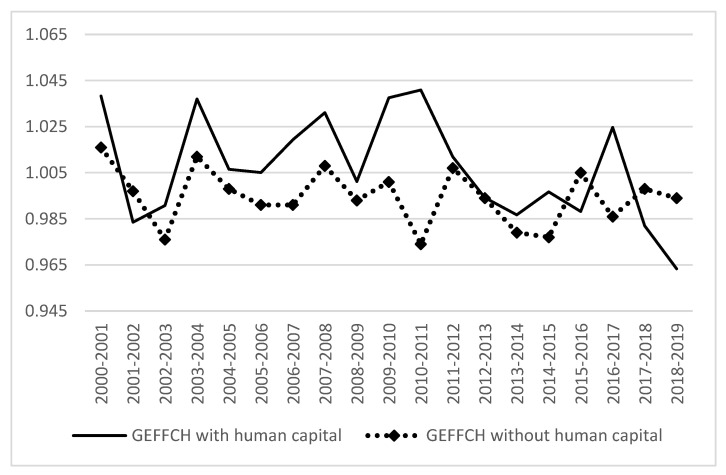
Comparison of technical efficiency decomposition.

**Figure 5 ijerph-19-13563-f005:**
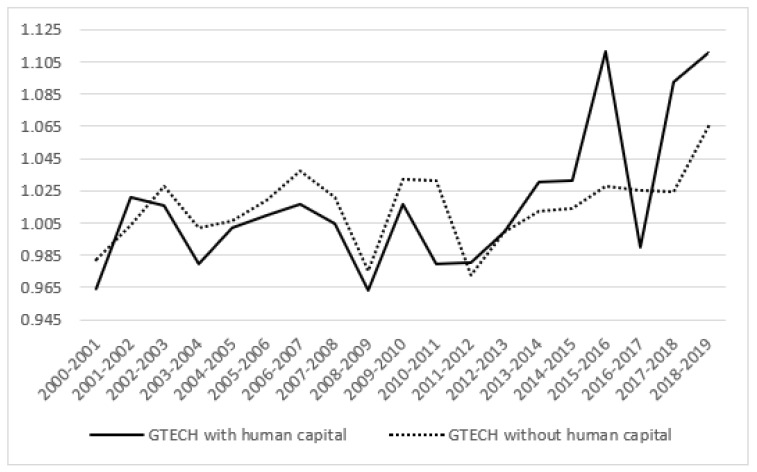
Comparison of technological progress decomposition.

**Figure 6 ijerph-19-13563-f006:**
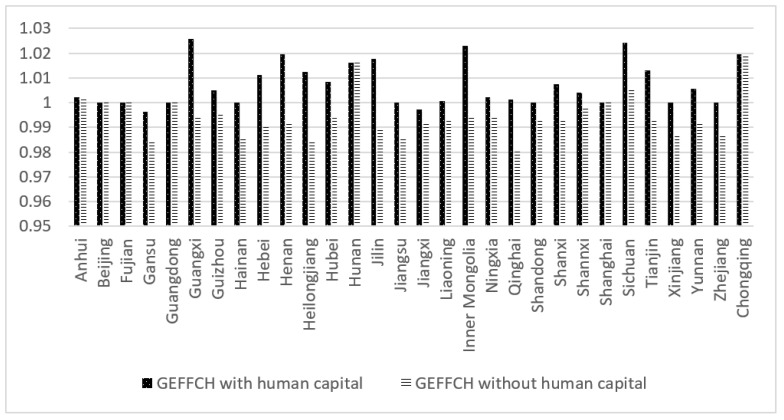
Comparison of technical efficiency among provinces and cities.

**Figure 7 ijerph-19-13563-f007:**
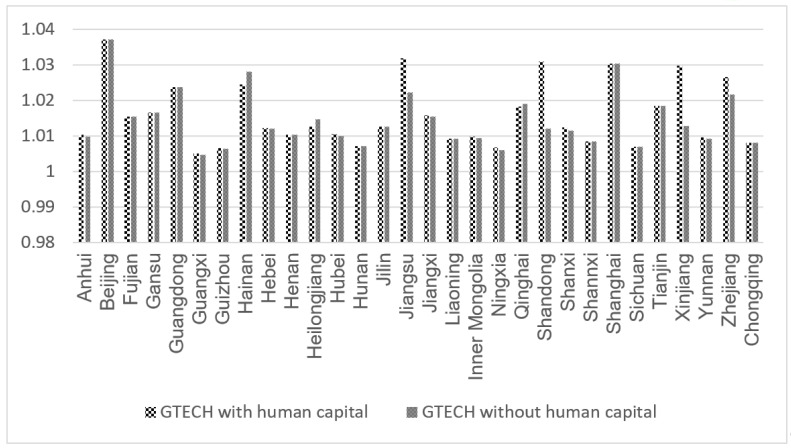
Comparison of technological progress among regions.

**Table 1 ijerph-19-13563-t001:** Cumulative years of education.

Level	Education Level	Years of Education	Cumulative Years of Education
1	Illiterate	0	0
2	Primary school	6	6
3	Junior high school	3	9
4	High school and secondary vocational school	3	12
5	Senior college	3	15
6	Undergraduate	4	16
7	Postgraduate and above	3	20

**Table 2 ijerph-19-13563-t002:** Input-output index description.

First Grade Indicators	Second Grade Indicators	Indicator Descriptions
Input	Education	Average years of education
	Health	The proportion of total health expenditure in GDP
	Scientific research	R&D personnel full-time equivalent
	Training	1.5% of the employee’s gross salary
	Capital	Fixed capital stock
	Labor quantity	Number of employees
	Energy	Total energy consumption
Desired output	GDP	Real GDP of each region based on the year 1978
	COD emissions	Including domestic and industrial pollution
Undesired output	SO_2_ emissions	
	Solid waste productions	

**Table 3 ijerph-19-13563-t003:** Descriptive statistics for variables.

Variable	Mean	Med	Sd	Min	Max
Education (year)	9.2066	9.0999	1.3249	5.934	13.997
Health (%)	5.4136	5.075	1.6688	2.64	11.97
Scientific research (ten thousand people a year)	84,553.155	47,506.5	109,321.5902	848	803,208
Training (hundred million yuan)	5.4754	4.0797	3.4439	0.7463	16.0052
Capital (hundred million yuan)	9030.689	5472.316	10,424.26	212.876	71,974
Labor quantity (ten thousand people)	2543.0582	2060.95	1696.5404	275.5	7072.625
Energy (tons of standard coal)	11,517.5554	9292.5	8155.3163	480	41,390
GDP (hundred million yuan)	15,300.477	10,093.16	16,753.87	263.68	110,184.54
COD emissions (ten thousand tons)	49.8761	39.44669	37.32154	1.9677	198.2
SO_2_ emissions (ten thousand tons)	62.6716	53.42	44.5431	0.19	200.3
Solid waste productions (ten thousand tons)	7916.4448	5468	8218.0186	75	52,037

**Table 4 ijerph-19-13563-t004:** Regional GTFP and its decomposition.

Region	Province	GEFFCH	GTECH	GTFP
Eastern	Beijing	1.0000	1.0371	1.0371
	Tianjin	1.0129	1.0184	1.0315
	Hebei	1.0111	1.0123	1.0235
	Liaoning	1.0005	1.0092	1.0097
	Shanghai	1.0000	1.0304	1.0304
	Jiangsu	1.0000	1.0318	1.0318
	Zhejiang	1.0000	1.0266	1.0266
	Fujian	1.0000	1.0155	1.0155
	Shandong	1.0000	1.0309	1.0309
	Guangdong	1.0000	1.0237	1.0237
	Guangxi	1.0259	1.0051	1.0311
	Hainan	1.0000	1.0246	1.0246
Central	Shanxi	1.0074	1.0125	1.0200
	Inner Mongolia	1.0231	1.0097	1.0330
	Jilin	1.0177	1.0126	1.0305
	Heilongjiang	1.0123	1.0126	1.0250
	Anhui	1.0023	1.0103	1.0126
	Jiangxi	0.9973	1.0158	1.0131
	Henan	1.0197	1.0104	1.0303
	Hubei	1.0083	1.0105	1.0189
	Hunan	1.0162	1.0071	1.0235
Western	Chongqing	1.0197	1.0081	1.0280
	Sichuan	1.0241	1.0070	1.0313
	Guizhou	1.0051	1.0066	1.0116
	Yunnan	1.0056	1.0096	1.0152
	Shaanxi	1.0042	1.0085	1.0127
	Gansu	0.9964	1.0166	1.0129
	Ningxia	1.0021	1.0068	1.0089
	Qinghai	1.0012	1.0183	1.0196
	Xinjiang	1.0000	1.0299	1.0299

**Table 5 ijerph-19-13563-t005:** Comparison of regional GTFP.

Region	GTFP with Human Capital	GTFP without Human Capital
Eastern	1.0263	1.0128
Central	1.0230	1.0062
Western	1.0189	1.0056

**Table 6 ijerph-19-13563-t006:** Decomposition of regional GTFP.

Region	Including Human Capital		without Human Capital	
	Efficiency Variation	Technology Progress	Efficiency Variation	Technology Progress
National	1.0071	1.0159	0.994	1.014
Eastern	1.0042	1.0221	0.9934	1.0196
Central	1.0116	1.0113	0.9951	1.0112
Western	1.0065	1.0124	0.9952	1.0104

## Data Availability

The data can be found at http://www.stats.gov.cn/tjsj./ndsj/ (accessed on 16 March 2022) and https://data.cnki.net/yearbook/Navi?type=type&code=A (accessed on 22 March 2022). The datasets used and/or analyzed during the current study are available from the corresponding author on reasonable request.

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
