# Peer review of "Measurement of China’s Green Total Factor Productivity Introducing Human Capital Composition"

_ijerph, 2022, doi:10.3390/ijerph192013563_

Round 1
Reviewer 1 Report
It is a complete and interesting issue. The paper introduces human capital and re-measures China’s green total factor productivity and its decomposition of 30 provinces from 2000 to 2019. It provides a new perspective to assess the relationship between human capital and GTFP, and provides guidance for China's green and sustainable development. However, heading 3 should be titled "Method Description".
Reviewer 2 Report
This is a very interesting paper on an interesting subject, but the presentation makes it impossible to review. There are many problems:
1. Acronyms mentioned without introduction - e.g., GML and SBM line 11; GDP line 111; DEA-ESDA in line 138; EPS in line 339, and there are more.
2. Mathematical development is hard to follow in lines 167-179. Mathematical development from line 180-188 is clean and easy to follow, as are lines 200-203 but lines 205 to 209 and lines 227-234 are hard to read again. Define "sack" variable in line 208.
Line 297, Equation (6) and the definitions of its parameters (lines 298-303) are completely missing.
3. These sections need more clear references - lines 50-55; lines 75-80, Schultz is not listed in the reference list (line 82); Solow (line 104) should have the year of publication; and there are more examples.
4. Table 3 does not divide the country into east and west and so lines 351-354 are not justified.
5. Figure 3a does not have the complete city or province name as Figure 3b mostly does.
6. Figure 5 needs a solid and a dotted line to distinguish between the curves.
7. Lines 585 to 630 are not complete.
8. The reference list is incomplete.
The above list gives examples of manuscript deficiencies but is not complete. The entire manuscript needs a comprehensive editorial rewrite. The data needs to be presented in a clearer manner.
Reviewer 3 Report
The paper presents an analysis of China's Green Total Factor Productivity based on human capital. The reading of the document is dense and not very fluid. It took me a long time to finish reading it and many parts I had to reread several times to understand what it meant.
I think it is necessary to review the wording and grammar of the text. There are redundant words, concepts that are repeatedly repeated in the same paragraph, excessively long sentences that make reading difficult, etc.
Also, not all acronyms are defined in the text and this makes it difficult to understand the document.
Some aspects of the document are not completed (such as the type of document, conflict of interest, funding,...)
The structure of the document should be reviewed. For example, the results section explains the method to use when it should explain the results obtained. The authors provide a lot of information but being poorly structured and very condensed it is easy to get lost.
Figure 3.a and 3.b cannot be compared separately. It would be interesting to display the data in a single graph for comparison.
There are phrases and information that are constantly repeated in the text over and over again (for example, the last sentence of section 1 and 2 or "provincial (cities, autonomous regions) appears 2 times in the same sentence on page 16 and 3 times more on the same page"). If the information was not constantly repeated, the text could be reduced and the speech would be clearer.
Reviewer 4 Report
This paper uses the Global Malmquist-Luenberger (GML) index model and human capital to re-measure GTFP and its decomposition in China. It is an interesting case study helping to achieve the goals of green development and a high-quality economy. However, this paper has two serious shortcomings. First, the logic of the whole paper is confusing. The introduction, materials, methods, and results are mixed. This article is lacking discussion part. Second, all input data came from open access sources, such as China Statistics Yearbook, but without data cleaning, data validation, and data sharing. Therefore, it would be a black box for the readers, and the readers could not believe the reality of the result and repeat the experiment (or model calculation).
All in all, this manuscript is not seemed to be a scientific article. Currently, this manuscript is still a report. I think this paper should be rejected. I suggest that the author should regroup all parts of the paper to improve the logic, and then resubmit it.
Some suggestions are following.
1. Line 32 CPC --> Communist Party of China (CPC)
2. Introduction part. Only one reference for all introductions? All cites from other paper need reference.
3. Introduction part. Introduce more details on GTFP and the GML model, such as some case studies from other countries.
4. The format of the reference, i.e. (author name and year), is error for MDPI.
5. Materials and Methods part. The manuscript describes the backgrounds of human capital and GTFP. It is not Materials and Method. But the authors put the Materials and Methods in the results part.
6. I cannot find the aims and objectives in the Introduction part.
7. Part 4.2 data source. All data are from open access sources, such as China Statistics Yearbook. But the authors did not describe how to extract and clean these data. So it is hard to believe the accuracy and reliability of the input data for readers. The authors should at least share these input data to the supplementary materials and Data Availability Statemen part. Without the source input data, the readers could not repeat the experiment by the GML model. Reproducibility of experiments is a fundamental principle of scientific research.
8. The quality of the English fluctuates throughout the manuscript. Overall the writing is very clear, but some sections suddenly contain awkward sentences.
9. Lack of discussion part.
Round 2
Reviewer 3 Report
The document has significantly improved its wording. Now it is much easier to understand what the authors mean. However, I still have several comments on the document:
1.- Format aspects are still not respected. The type of paper is not indicated, the contribution of each author, there is an error in the affiliation of the authors, the format of the references does not coincide with that of the journal template, etc. It may seem like a minor aspect but I consider it necessary out of deference to the entity, the editor and the reviewers.
2.- A minor aspect is the typographical errors that exist in the text. For example, in the introduction it is indicated that the analyzed data is from 2000-2009 instead of 2000-2019.
3.- The length of the work is high and is not consistent with the amount of information provided. Let me explain: the work recurrently repeats the same ideas over and over again. By constantly pivoting on the same idea for 17 pages, the reader is discouraged from continuing to read.
4.- Figure 1 and Table 4 do not show the same data? There is no point in duplicating information.
5.- It seems obvious that when using other data to calculate an index, the result of the index will be different. Therefore, if it is taken as a starting hypothesis, it will always be true. In that sense, I believe that the contribution of the authors is limited.
6.- Without the intention of continuing to increase the number of pages of the document, I miss a critical analysis of the data and its results. The authors, for example, do not include the limitations of the study they present. Are there no limitations? Has the data you propose to include been measured in the same way or with the same criteria for 20 years in China?
7.- I have serious doubts about the value of the statistical variables (Table 3). In what terms are they measured? For example, is scientific production measured in quantity or quality? what criteria is followed? The energy is consumed? or is it about energy efficiency? Have these parameters always been measured the same?
Reviewer 4 Report
The revised version has been improved a lot.
My comment is that this paper could be accepted.
Author Response
Thanks to the reviewer for the approval of the revised version and comment on receiving this paper.